# Laboratory Investigation of Rubberized Asphalt Using High-Content Rubber Powder

**DOI:** 10.3390/ma13194437

**Published:** 2020-10-06

**Authors:** Guoqing Wang, Xinqiang Wang, Songtao Lv, Lusheng Qin, Xinghai Peng

**Affiliations:** 1School of Civil and Transportation Engineering, Hebei University of Technology, Tianjin 300401, China; wanggq98@163.com; 2Hebei Transportation Investment Group Corporation, Shijiazhuang 050091, China; hbglzc@163.com; 3National Engineering Laboratory of Highway Maintenance Technology, Changsha University of Science & Technology, Changsha 410004, China; lst@csust.edu.cn (S.L.); pengxinghsi@stu.csust.edu.cn (X.P.)

**Keywords:** high-content rubberized asphalt (HCRA), physical performance, rheological properties, microscopic characteristics

## Abstract

Rubberized asphalt (RA) has been successfully applied in road engineering due to its excellent performance; however, the most widely used rubber content is about 20%.To improve the content of waste rubber and ensure its performance, seven rubberized asphalts with different powder content were prepared by high-speed shearing. Firstly, penetration, softening point, and ductility tests were carried out to investigate the conventional physical features of high-content rubberized asphalt (HCRA). Then, the dynamic shear rheometer test (DSR) was conducted to estimate the high-temperature rheological properties. The bending beam rheometer test (BBR) was carried out to evaluate the low-temperature rheological performance. Finally, combined with the macroscopic performance test, the modification mechanism was revealed by the Fourier transform infrared reflection (FTIR) test, and scanning electron microscope (SEM) analysis was used to observe the microscopic appearance before and after aging. The results show that rubberized asphalt has excellent properties in high- and low-temperature conditions, and fatigue resistance is also outstanding compared with neat asphalt. As the crumb rubber content increases, it is evident that the 40% RA performance is the best. The low-temperature properties of HCRA are better than the traditional 20% rubberized asphalt. This study provides a full test foundation for the efficient utilization of HCRA in road engineering.

## 1. Introduction

Road rubberized asphalt (RA) is made by the high-temperature shear mixing of asphalt, waste tire rubber powder, and various admixtures [1]. This not only recognizes the recycling of waste tire rubber but also improves the properties of neat asphalt [2]. Rubberized asphalt has always been a research hotspot in the pavement industry [3].

The entire development process of rubberized asphalt can be divided into four stages according to the different application methods in different periods [4]. The first stage is mixing rubber powder and aggregate and then adding asphalt to produce a rubber-modified asphalt mixture [5,6]. The second stage is mixing asphalt + rubber powder + rubber oil, stirring in a tank at a temperature of 180 to 200 °C for about 40 to 60 min, then mixing with aggregate to produce a rubber-modified asphalt mixture [7]. The third stage involves factory-based stabilized rubberized asphalt, asphalt + rubber powder, stirred at the specified temperature and time and then added to the product tank when the product standard is reached [2,8]. Thermal storage does not stratify the mixture, and it does not decay. It can be used in a variety of mixture structures. After environmental protection oktreatment, the fourth stage is where the rubber powder is mixed with asphalt, stirred at the specified temperature and time, then added to the product tank when it reaches the product standard, at which point the tank is sealed for sampling inspection, and then it leaves the factory after quality control is completed [9]. It is suitable for all types of mixtures, and the used method and environment are similar to styrene–butadiene–styrene (SBS)-modified asphalt [2,10].

The preparation of rubberized asphalt is divided into the dry method and the wet method [10,11,12]. The dry method involves mixing aggregate and rubber powder first and then adding asphalt for wet mixing. In this method, the contact area and sufficient reaction time between rubber powder and asphalt are insufficient. The performance of neat asphalt cannot be improved at all by adsorbing light components. Therefore, the dry method improves the asphalt mixture’s performance, meaning that it cannot be regarded as modified asphalt in a real sense [13].

The wet method process of rubberized asphalt involves heating the neat asphalt to a certain temperature, adding rubber powder and stirring until it is wholly swelled, achieving uniform dispersion through high-speed stirring, shearing, or grinding [14]. At present, thermoplastic rubbers such as SBS and styrene–isoprene–styrene (SIS), and thermoplastic resin modifiers such as ethylene-vinyl acetate copolymer (EVA) and polyethylene (PE) have low compatibility with asphalt [3]. Simple mechanical stirring takes a long time, and the effect is not outstanding [15]. Such modifiers must use colloid mills or special shearing equipment to ensure that the modifiers are fully dispersed in the neat asphalt [16,17].

At present, traditional rubberized asphalt has the following shortcomings during use: it is easily degraded at high temperatures and cannot be stored for a long time [18,19]. Rubberized asphalt produced on-site is time limited; after a certain point, its performance will be significantly reduced—e.g., in a low construction environment [8,13]. Most of the manufacturing process of asphalt rubber is on-site production, and a sufficient working surface must be added at the construction site to complete the production of asphalt rubber [19,20]. If the project volume is not large, there is the risk of producing a large amount of waste, which is also an obstacle to the promotion of rubberized asphalt—underdeveloped construction and workability [3,7,15]. Because rubber-modified asphalt’s viscosity is too large, the workability of construction is a deviation, and the amount of rubber powder is not high [16,21].

Environmentally stable rubberized asphalt modifiers are new types of composite materials [11,22,23]. They are based on an appropriate grade of industrial and recycled polymers as their primary and auxiliary materials, using self-developed rubber molecular chain finishing, precise molecular weight control, and stable mixed reaction technology [1,3,24]. Their primary mechanism is desulfurization, which reduces the molecular weight of rubber powder [1,25]. After environmental treatment, the rubberized powder has excellent compatibility with asphalt, the production efficiency is greatly improved, and it is easy to meet the needs of different scale mixing plants.

The objective of this research is to improve the rubber powder content in an environmentally stable way. To achieve this, the conventional physical properties and the high- and low-temperature rheological properties of the prepared rubberized asphalt were tested to study its macroscopic physical and mechanical properties. Going one step further, the Fourier transform infrared reflection (FTIR) test and scanning electron microscope (SEM) analysis were carried out to analyze its microscopic characteristics. This research provides an experimental basis for the application of high-content rubberized asphalt in road engineering and lays a solid foundation for the customized design of rubberized asphalt mixture.

## 2. Preparation of High-Content Rubberized Asphalt

### 2.1. Raw Materials

#### 2.1.1. Neat Asphalt

Using 70# neat asphalt, we referred to the “Standard Test Methods of Bitumen and Bituminous Mixture for Highway Engineering” (JTG E20-2011) [26] to determine the major indices of asphalt. The performance test is shown in Table 1.

#### 2.1.2. Rubber Powder

The particle size of the waste rubber powder was less than 0.6 mm, and the conventional properties are shown in Table 2.

### 2.2. Preparation of Rubberized Asphalt

According to a specific method, the stabilized rubberized asphalt was prepared [27], in which the blending content of waste tire rubber powder was 20%, 25%, 30%, 35%, 40%, 45%, and 50% (internal blending),this is a total of seven types of rubberized asphalt. Rubberized asphalt with more than 20% rubber powder was defined as a high-content rubberized asphalt (HCRA). Processing equipment, as shown in Figure 1, and the preparation steps are as follows:(1)Weigh the raw materials of the modified asphalt according to the mass ratio.(2)Heat the neat asphalt to 160 °C, and then transport it to the pilot reactor.(3)Add viscosity reducing agent and reinforcing agent to the pilot reactor, and stir evenly. The viscosity reducing agent is a mix of activator 950, dioctenyl phthalate, and epoxy fatty acid methyl ester. The reinforcing agent is natural asphalt, petroleum resin, and phenolic resin.(4)Add different contents of rubber powder into the pilot reactor, mix and stir, control the temperature at 180–190 °C, and allow a swelling time of 50 min.(5)Pump the aforementioned asphalt mixture into a high-power colloid mill for vigorous shear grinding; the high-power colloid mill performs strong shear grinding at a speed of 2000 rpm and a shear time of 15 min.(6)Add stabilizer to the asphalt mixture after vigorous shear grinding. The stabilizer is a mixture of sulfur and organic sulfide.(7)Pump the combined asphalt mixture and stabilizer to a colloid mill for weak shear grinding to obtain a modified asphalt product (2000 rpm speed and a shear time of 10 min).

## 3. Experiment Methods

In this work, all macroscopic performance tests were carried out in accordance with the specifications of the “Standard Test Methods of Bitumen and Bituminous Mixture for Highway Engineering” (JTG E20-2011) [26].

### 3.1. Penetration, Ductility, Softening Point, and Rotation Viscosity

The penetration, ductility, softening point, and rotation viscosity of neat asphalt and rubberized asphalt are measured. The standard conditions for penetration tests are a temperature of 25 °C, a penetration time of 5 s, a 100 g load, and the same sample is tested in parallel three times. The ductility test temperature is 25 °C, and the tensile speed is 5 ± 0.25 cm /min, and the same sample is tested in parallel three times. The softening point of neat asphalt and rubberized asphalt is measured by the ring and ball method, and the same sample is tested twice in parallel. Rotation viscosity was carried out by Brookfield at 180 °C, and the same sample was tested in parallel three times.

### 3.2. Dynamic Shear Rheology Test (DSR)

For the original asphalt test, the sample diameter is 25 mm, and the thickness is 1 mm. Dynamic shear rheometer (DSR) equipment and strain control mode are used to test the specimen, DSR equipment is shown in Figure 2. For this, we apply a sine oscillating load with a frequency of 0.1–100 rad/s to test the neat asphalt and rubberized asphalt’s rheological properties to determine their dynamic modulus and phase angle. The frequency sweep temperature is 60 °C.

The specific test steps are as follows:(1)Prepare samples according to standard methods.(2)Select a test plate with a diameter of 25 ± 0.05 mm and clean its surface. Place it on the testing machine and move the top plate to make the plate gap 1 ± 0.05 mm.(3)Take out the test board, pour the sample on the test board, and install the test board on the testing machine after the sample hardens. Moreover, move the test board to squeeze the sample, heat the test piece repairer, and clean up the overflowing test piece. Then, adjust the gap to 1 ± 0.05 mm.(4)After the temperature control box temperature remains stable at 60 °C for 2 min, start loading, and perform frequency scanning.

### 3.3. Bending Beam Bheometer Test (BBR)

After short-term aging (rolling thin-film oven test (RTFOT) 85 min) and long-term aging (pressurized aging vessel (PAV) 20 h), the flexural creep stiffness and creep rate of neat asphalt and rubberized asphalt are measured by a bending beam rheometer at −12, −18 and −24 °C. Three parallel tests are conducted for each sample and temperature. BBR equipment is shown in Figure 3.

The specific test steps are as follows:(1)Put the test piece into the prepared thermostatic bath immediately after demolding, keep it there for 60 ± 5 min, then place it on the support, and keep the thermostatic bath within ±0.1 °C of the test temperature;(2)Input relevant information such as test temperature, test load, and test piece data into the computer;(3)Manually apply a contact load of 35 ± 10 mN to the specimen, and ensure that the application time does not exceed 10 s. The specimen must be in contact with the load head during the application process;(4)Activate the automatic loading system, apply an initial load of 980 ± 50 mN within 1 ± 0.1 s, reduce the load to 35 ± 10 mN, and maintain it for 20 ± 1 s. Apply a test load of 980 ± 50 mN for 240 s, and the computer will automatically record and calculate the load and deformation values from 0.5 s at intervals of 0.5 s. Remove the test load and return the system to a contact load of 35 ± 10 mN, remove the test piece, and proceed to the next test.

### 3.4. Aging Test

A rolling thin-film oven test (RTFOT) and an accelerated aging test of the asphalt binder using a pressurized aging vessel (PAV) are used to simulate short-term aging and long-term aging, respectively.

#### 3.4.1. Rolling Thin-Film Oven Test (RTFOT)

Short-term aging test procedure: weigh 35 ± 0.5 g asphalt sample and place it in a short-term aging bottle; adjust the rotating oven to a certain level, and preheat it to 163 ± 0.5 °C for no less than 16 h to heat the air in the box evenly. Adjust the temperature controller and put all the sample bottles into the metal ring rack. At this time, the oven temperature should reach 163 ± 0.5 °C within 10 min; adjust the distance between the air nozzle and the opening of the sample bottle to 6.35 mm. Furthermore, to adjust the flow rate, the historical air flow rate is 4000 ± 200 mL/min; the test’s total duration is 85 min. This kind of equipment is produced by China Beijing Zhongjian Road Industry Instrument and Equipment Co., Ltd., and the equipment is shown in Figure 4a.

#### 3.4.2. Accelerated Aging Test of Asphalt Binder Using a Pressurized Aging Vessel (PAV)

This kind of equipment is produced by China Beijing Zhongjian Road Industry Instrument and Equipment Co., Ltd., and the equipment is shown in Figure 4b. Long-term aging test procedure:(1)Pour the asphalt residue from the rolling thin-film oven test into the container.(2)Balance the standard film oven test sample tray of known quality, add 50 ± 0.5 g of asphalt to the plate, and make the asphalt film thickness about 3.2 mm.(3)Put the tray rack in the pressure vessel, select the temperature of the pressure aging container, then turn on the heater to preheat the rack to the selected aging temperature of 100 °C.(4)After reaching the aging temperature, quickly open it and put it into the prepared sample tray, then and close the pressure vessel.(5)When the internal temperature of the pressure vessel is lower than 2 °C (required within 2 h), start to supply air of 2.1 ± 0.1 MPa pressure and time.(6)Keep the temperature and air pressure constant for 20 h ± 10 min; after 20 h, open the pressure reducing valve to make the pressure in the pressure vessel reach the same pressure as the outside pressure in 8–15 min.

### 3.5. Fourier Transform Infrared Reflection (FTIR)

In order to reveal the mechanism of rubberized asphalt, infrared spectroscopy is used for functional group analysis. Fourier transform infrared reflection (FTIR) is an analysis method to obtain the molecular composition of a substance based on the absorption of infrared light by a substance. Usually, infrared reflection adopts a mid-infrared band, which ranges from 4000–500 cm^−1^, among which 4000–1300 cm^−1^ is the functional group and 1300–600 cm^−1^ is the fingerprint area. The former is an infrared spectrum to analyze the most valuable analysis area. While the latter is involved, slight differences in molecular structure will produce changes [9,16,28].

The neat asphalt and rubberized asphalt are measured by the nexus Fourier transform infrared spectrometer produced by Thermo Nicole (Wisconsin, WI, USA). The specific steps are as follows:(1)Use carbon tetrachloride (CCI_4_) reagent to fully dissolve the modified asphalt (0.1 g asphalt needs 2 mL of CCI_4_ reagent to dissolve).(2)After dissolving completely, drop 2 drops on a potassium bromide (KBr) wafer and air dry.(3)When the sample is cooled, it can be put into the sample tank for scanning. During the test, the beam acquisition interval is set at 400–4000 cm^−1^, the scanning times are 32, and the resolution is 4 cm^−1^.

### 3.6. Scanning Electron Microscope (SEM)

The information contained in the electronic scanning image can well reflect the surface morphology of the sample. To observe the microscopic morphology of the neat asphalt and rubberized asphalt, the asphalt sample is imaged by a scanning electron microscope, and the magnification is 200 times. This research chooses the S-3400N tungsten filament scanning electron microscope produced by Hitachi (Tokyo, Japan), which guarantees a resolution of 10 nm at a low acceleration voltage of 3 kV. In order to obtain better scanning electron microscopy images, sputtering ion equipment is used to spray the sample gold before scanning

## 4. Test Results and Discussions

### 4.1. General Physical Properties

The test results of the three major indices of rubberized asphalt with different rubber powder contents are shown in Figure 5. Figure 5a shows the penetration of rubberized asphalt; Figure 5b shows the ductility of rubberized asphalt; Figure 5c shows the softening point of rubberized asphalt.

Penetration, ductility, and softening point tests were carried out for each content of rubber-modified asphalt. It can be obtained from Figure 5a that the penetration of rubberized asphalt with different contents of rubber powder occurs in different zones. The penetration of 20% rubberized asphalt is in the range of 40–60 (0.1 mm). The penetration of 25–40% rubberized asphalt is concentrated in the range of 60–80 (0.1 mm). The penetration of 45–50% rubberized asphalt is distributed in the range of 80–90 (0.1 mm). Thus, as the rubber powder content increased, the penetration increased.

Figure 5b demonstrates that the ductility of rubberized asphalt meets the requirements of “Asphalt rubber for highway engineering” (JT/T798-2019) in a cold area greater than 100 mm [29]. As the content of rubber powder increases, the ductility first increases and then decreases. Among them, the rubberized asphalt with 35% rubber powder has the highest ductility, reaching 315.7 mm.

It is obtained from Figure 5c that the softening point of rubberized asphalt mixtures with different contents meets the standard requirements, which is mostly concentrated at 65–73 °C. They are 31.7–45.7% higher than neat asphalt, indicating that rubber powder can effectively improve the high-temperature properties of neat asphalt [16]. However, there is no apparent difference in the softening point of rubberized asphalt in various rubber powder contents. This cannot effectively distinguish the difference in the high-temperature performance and viscoelastic properties of rubberized asphalt [20].

It can be seen from Figure 5d that the viscosity of all rubberized asphalt forms at 180 °C is less than 3 Pa·s, which has an excellent construction mixing performance. Compared with 20% rubberized asphalt, the viscosity of HCRA is slightly reduced at 25%. When the content is more than 25%, the viscosity is greater than 20% RA, and the viscosity of 50% RA is 127.17% higher than 20% RA.

### 4.2. High-Temperature Rheological Properties

Under the condition of 60 °C, the frequencies of eight kinds of asphalt were scanned by DSR equipment. The complex shear modulus and phase angle were measured at different rates, and the rutting index and fatigue index were calculated. The rutting index in high-temperature conditions and the fatigue index of various asphalts were compared.

#### 4.2.1. Viscoelastic Properties

Figure 6 manifests that the complex shear modulus and phase angle of the asphalt is affected by the loading frequency. As the loading frequency increases, the complex shear modulus increases, and the phase angle decreases.

It can be found, from Figure 6a, that the dynamic shear modulus of 40% RA is the largest, and the neat asphalt is the smallest. The complex shear modulus is not only influenced by the rubber powder content, but also by comprehensive factors such as rubber powder content and additives [25,30]. Figure 6a represents that, at 10 Hz, the dynamic shear modulus of 25% RA increases by 11.6%, and that of 40% RA increases by 145.3%, and the dynamic shear modulus of the other rubberized asphalt lies between them, while the dynamic modulus of the neat asphalt is less than 14 kPa.

Figure 6b manifests that the phase angle of the neat asphalt is concentrated between 80° and 90°, and the phase angle of rubberized asphalt is concentrated between 50° and 65°, among which the phase angle of rubberized asphalt with 20–30% rubber powder is at the same level. Moreover, 40% RA, 45% RA, and 50% RA are at the same level. Generally speaking, with the increase in rubber powder, the phase angle shows a decreased trend [31,32]. Figure 6b summarizes that the phase angle of rubberized asphalt is much smaller than the neat asphalt. Consequently, rubber powder plays a crucial role in modification due to the swelling of rubber powder and the formation of a three-dimensional network structure with asphalt, which increases the flow resistance.

The storage modulus and loss modulus of rubberized asphalt is further analyzed. As shown in Figure 7, Figure 7a shows the storage modulus of rubberized asphalt, and Figure 7b shows the loss modulus of rubberized asphalt.

It can be seen from Figure 7 that no matter the storage modulus or loss modulus, 40% RA is the highest, neat asphalt is the lowest, and another asphalt type lies between them, which is consistent with the conclusions of Figure 5.

#### 4.2.2. Anti-Rutting Performance

According to the complex shear modulus and phase angle in Figure 6, the rutting index of asphalt is calculated by $G^{*}/\sin\delta$ (Here, the $G^{*}$ is the complex shear modulus, and the δ is the phase angle.) and three frequencies of 0.1, 1, and 10 Hz are selected [33]. The rutting index of asphalt is shown in Figure 8. The more significant the rutting index, the more preferable the high-temperature stability of asphalt is and the better its resistance to permanent deformation is.

Figure 8 indicates that the rutting index of asphalt is comprehensively affected by the loading frequency and the content of rubber powder. As the frequency increases, the rutting index of asphalt shows an increasing trend. When the frequency is 1 and 10 Hz, the rutting factor of neat asphalt is seven times and 42 times 0.1 Hz.

The addition of rubber powder can effectively improve the rutting resistance of neat asphalt and improve the anti-rutting performance at high temperature. The reason is that the interaction between rubber powder and neat asphalt is apparent, forming a physical crosslinking effect [24]. This makes the proportion of asphalt components change proportionally, forming a stable gel structure, thereby improving the high-temperature properties of neat asphalt [33]. A rubber powder content greater than 20% is defined as high-content rubberized asphalt. When the rubber powder content is greater than 25%, as the rubber powder content increases, the rutting index first increases and then decreases. When the rubber powder content is more than 35%, the high-temperature properties of rubberized asphalt with high content are better than those of 20% rubberized asphalt.

When comparing Figure 8 and Figure 5, there is no visible difference in the softening point of various asphalts, but there are remarkable differences in the rutting index. Dynamic shear rheology can profoundly analyze the high-temperature performance of asphalt and determine the influence of frequency on it.

#### 4.2.3. Fatigue Resistance

According to the complex shear modulus and phase angle in Figure 6, the fatigue index of asphalt is calculated by *G***sinδ*, taking the asphalt fatigue index in 0.1, 1, and 10 Hz as an example, as shown in Figure 9 [15].

Compared with Figure 8 and Figure 9, this demonstrates that the asphalt fatigue index’s variation pattern is consistent with the rutting index, which increases with the loading frequency. The fatigue index of rubberized asphalt is higher than the neat asphalt. At three frequencies of 0.1, 1, and 10 Hz, the fatigue index of rubberized asphalt with 20% rubber powder is increased by 5.8, 2.9, and 1.9 times, respectively; the difference is within one order of magnitude. For high-content rubberized asphalt, the fatigue index increases first and then decreases with the increase in rubber powder content from 25%. The fatigue index of rubberized asphalt with 40% rubber powder is the largest, and the overall fatigue index is relatively stable and remains at a low level.

### 4.3. Low-Temperature Rheological Properties

The BBR test results of eight kinds of asphalt are shown in Figure 10, in which Figure 10a is the stiffness modulus (*S*), and Figure 10b is the creep rate (*m*) [11,23]. The samples have all been aged by RTFOT and PAV [24,34].

The test results in Figure 10a show that no matter what kind of asphalt is used, the stiffness modulus will increase with the decrease in temperature, which is in line with the actual pavement situation. Under the three test temperature conditions, the low-temperature stiffness modulus of rubberized asphalt is smaller than the neat asphalt. The stiffness modulus is smaller as the increase in rubber powder content, and the cracking resistance in the low-temperature environment of rubberized asphalt improves. Compared with the rubberized asphalt with 20% rubber powder, the low-temperature properties of high-content rubberized asphalt do not show a downward trend. On the contrary, with the increase in the rubber powder content, the low-temperature modification effect is more significant.

Figure 10b illustrates that asphalt’s creep rate will decrease when the temperature is lowered; accordingly, cracking is more likely to occur [3,22]. Under the three temperature conditions, rubberized asphalt’s creep rate is much better than the neat asphalt; compared with rubberized asphalt with 20% rubber powder, the creep rate of high-content rubberized asphalt is higher than it. Unlike the stiffness modulus, the creep rate does not always increase with the rubber powder content. At ‒24 °C, The creep rate of rubberized asphalt with 20% and 25% powder content is extremely equivalent.

Referring to the limit requirements of stiffness modulus *S* ≤ 300 mPa and creep rate *m* ≥ 0.3, the low-temperature grade of neat asphalt cannot reach −12°C, but the low-temperature grades of 20% RA, 25% RA, 30% RA, 35% RA, 40% RA, 45% RA and 50% RA are −12, −12, −18, −18, −18, −24, −24 °C, respectively [22,23].

Further analysis of the low-temperature sensitivity of asphalt manifests that the stiffness index can represent the low-temperature sensitivity of asphalt, and the calculation is as follows [3]:(1)$$\mathrm{lg}S=S_{TS}T+C$$

*S_TS_* is the stiffness index; *T* is the test temperature; *C* is the regression constant.

According to Equation (1), the logarithm of the stiffness modulus is linearly fitted with temperature. The fitting results are shown in Figure 11, and the fitting parameters are shown in Table 3.

Figure 11 and Table 3 demonstrate that with the rise in temperature, the stiffness modulus of rubberized asphalt descends linearly, and the fitting parameter *C* represents the intercept with the ordinate. It can be seen that the intercept of neat asphalt is the largest, that is, the line position is the highest. With the rise in rubber powder content, the line position gradually decreases. *S_TS_* implies the slope of the fitting curve, that is, the sensitivity to low temperature. From the fitting results and the slope of the curve, the sensitivity of neat asphalt to low temperature is small; 40% RA is the most sensitive to low temperature. Furthermore, the low-temperature stability of high-content rubberized asphalt is similar to the rubberized asphalt with 20% rubber powder, which indicates that high-content rubberized asphalt has an excellent anti-cracking performance. We can summarize that low-temperature properties are the primary benefit of the application of crumb rubber powder compared with existing works.

### 4.4. Fourier Transform Infrared Reflection (FTIR)

Fourier transform infrared reflection usually analyzes the chemical structure of petroleum asphalt and a polymer [9,16]. This is mainly carried out through the absorption of polymers under different wavelengths of infrared radiation. A slice of the polymer components absorbs the radiation of part of the wavelengths and weakens the infrared light, thus forming the infrared spectrum. The infrared spectrum of a substance is the reaction of its molecular structure, and it plays an essential role in identifying the particular functional groups in asphalt and polymers. The neat asphalt and rubberized asphalt with 30% rubber powder were tested by Fourier transform infrared reflection to reveal the modification mechanism. The test results are shown in Figure 12, the red and blue lines represent the transmittance of rubber asphalt (30%RA) and neat asphalt at different wave numbers, respectively.

It can be obtained from Figure 12 that the peak positions of rubberized asphalt and neat asphalt are the same, and there is no new absorption peak. The specific position of the wave crest has been identified in the figure. The results are indicative of the chemical functional groups of saturated alkanes and are mainly divided into C–H vibrations and C–C skeleton vibrations [30]. The C–H vibration includes a C–H stretching vibration (absorption peak in the range of 3000–2850 cm^−1^) and a C–H variable angle vibration (near 1460 and 1370 cm^−1^). In contrast, the absorption peak of C–C skeleton vibration occurs in the range of 1100–1020 cm^−1^. The stretching vibration absorption peak of the carbon–carbon double bond (C=C) mainly occurs in the range of 1700–1370 cm^−1^, while the absorption peak of the C=C stretching vibration of the aromatic ring mainly occurs in the range of 1610–1370 cm^−1^ due to the sizeable π-conjugated system [22]. The olefin (trans) C–H out-of-plane bending vibration band is relatively stable, mainly at 965 cm^−1^, and the carbonyl absorption peak mainly occurs in the interval of 1800–1650 cm^−1^ [9,28].

Figure 12 shows visible absorption peaks at 1372, 1455, 2853 and 2923 cm^−1^ in the infrared spectra of neat asphalt and rubberized asphalt samples [10]. Among them, 1372 and 1455 cm^−1^ belong to a methyl (–CH3) bending vibration. At the same time, the absorption peak at a 2853 cm^−1^ wavelength is caused by the C–H stretching vibration of alkanes, and the absorption peak at 2923 cm^−1^ gives rise to the stretching vibration of the methylene C–H bond [8].

For other absorption peaks, at a wavelength of 751 cm^−1^, these are attributed to the out-of-plane bending vibration of the olefin C–H [4]. At a wavelength of 807 cm^−1^, the absorption peak is triggered by the out-of-plane bending vibration of the olefin C–H. At an 868 cm^−1^ wavelength, this results from an out-of-plane bending vibration peak of the hydroxyl group (O–H) [14]. The absorption peak at the wavelength of 1606 cm^−1^ is attributed to the aromatic ring C=C stretching vibration. The absorption peak at the wavelength of 1739 cm^−1^ is set off by the aldehyde group (C=O stretch) vibration [1].

Comprehensive analysis confirms that the rubber powder and asphalt in the rubberized asphalt are physically blended, and there is no chemical reaction between them. Instead, the added stabilizer and viscosity reducer may have interacted. 

Similarly, as with SBS-modified asphalt, when the proportion of each component in asphalt is appropriate, SBS will interact with asphalt immediately after being evenly dispersed into asphalt, and then interact with asphalt. Thus, the swelling phenomenon occurs. Absorbed asphalt molecules surround an isolated SBS particle, creating an SBS molecular chain in the particle after full swelling. The incorporation of rubber powder improves the high- and low-temperature stability of the neat asphalt. The rubber powder and the neat asphalt form a physical cross-linking effect, leading to the proportions of the asphalt components changing to form a stable gel-type structure, thereby improving the high- and low-temperature performance and stability of neat asphalt [14].

### 4.5. Scanning Electron Microscope (SEM)

The neat asphalt, 20% rubberized asphalt, and 30% and 50% high-content rubberized asphalt were selected, and they were imaged by a scanning electron microscope with a magnification of 200 times. The results are shown in Figure 13.

It can be concluded from Figure 13 that, for original asphalt, the surface of the neat asphalt is relatively smooth, but there are apparent wrinkles or silver streaks [10]. The rubberized asphalt’s surface with 20% rubber powder is smooth and flat because the viscosity reducer plays a prominent role in it. Compared with the neat asphalt, there are fewer wrinkles. No rubber powder agglomeration phenomenon is found on the asphalt surface, which indicates that the rubber powder has excellent compatibility with the neat asphalt [30]. These results mean that the rubberized asphalt has promising stability, and the effect of the stabilizer is remarkable. When the rubber powder content is 30%, it is similar to that when the rubber powder content is 20%. There is no apparent change, which indicates that the increase in rubber powder content will not adversely affect the rubberized asphalt [11].

On the contrary, when the rubber powder content is as high as 50%, the rubberized asphalt surface gradually becomes rough and has an apparent striped shape [35]. This is because rubber powder is evenly distributed in the neat asphalt, and swelling occurs and forms a stable three-dimensional network junction with asphalt [15]. From this point of view, the advantages of the preparation method of high-content rubberized asphalt are explained.

For the asphalt after RTFOT and PAV, compared with the original asphalt, the neat asphalt has apparent cracks, the structure has been damaged, and its anti-aging performance is insufficient [35]. The most significant change after the aging of rubberized asphalt with 20% rubber powder is the surface folds—that is, the rubberized asphalt becomes more viscous; the 30% and 50% rubberized asphalt are similar. The difference is that the higher the content, the more pronounced the wrinkles are—that is, aging increases the rubberized asphalt’s flow resistance. The rubberized asphalt surface still has no agglomeration phenomenon and no structural damage [3], proving the superior aging resistance of high-content rubberized asphalt.

### 4.6. Economic Analysis

In order to further analyze the advantages of HCRA prepared in this study, the preparation costs of HCRA are briefly analyzed, and the specific calculations are shown in Table 4.

It can be seen from Table 4 that the cost of rubberized asphalt and HCRA is about 3400 RMB/ton, while the SBS-modified asphalt on the market is usually higher than 5000 RMB/ton in comparison, leading to significant economic benefits and market competitiveness [36]. Simultaneously, the HCRA prepared in this study has been successfully applied in actual pavement engineering. The application effect shows that the HCRA has an excellent road performance, consistent with the laboratory test results, and has broad application prospects.

## 5. Conclusions

In this work, high-content rubberized asphalt with a rubber content of up to 50% has been obtained through a specific preparation method. The high-content rubberized asphalt has been analyzed in depth from macroscopic and microscopic perspectives. The results indicate that the high-content rubberized asphalt has remarkable high- and low-temperature, fatigue resistance, and anti-aging properties, which provide an effective way to utilize waste rubber efficiently. The main conclusions are as follows:(1)No matter the conventional physical properties or rheological performance, high-content rubberized asphalt has excellent high-temperature properties, low-temperature characteristics, and fatigue resistance compared with neat asphalt.(2)Compared with the rubberized asphalt with 20% rubber powder, high-content rubberized asphalt’s performance is adjustable and controllable. With the rise in rubber powder content, the high-temperature properties first increase and then decrease, and the low-temperature characteristics are enhanced.(3)The FTIR test shows that the HCRA is physically modified. The rubber powder forms an excellent network connection with the neat asphalt, as confirmed by SEM imaging. The microscopic appearance of high-content rubberized asphalt before and after aging shows little change, and it still has favorable compatibility and stability after long-term aging.(4)In this research, a comprehensive test of high-content rubberized asphalt was carried out from macro and micro perspectives, highlighting the significance of applying high-content rubberized asphalt in practical engineering. The next examination should employ the Linear Amplitude Sweep (LAS) and Multiple Stress Creep Recovery (MSCR) tests, while also focusing on the road performance and mechanical properties of high-content rubberized asphalt mixtures.

## Figures and Tables

**Figure 1 materials-13-04437-f001:**
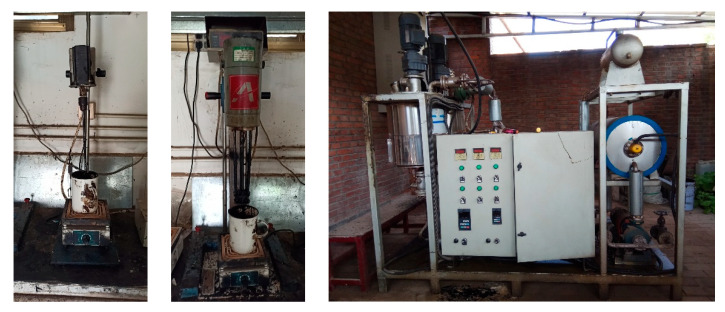
Equipment for high-content rubberized asphalt (HCRA) preparation.

**Figure 2 materials-13-04437-f002:**
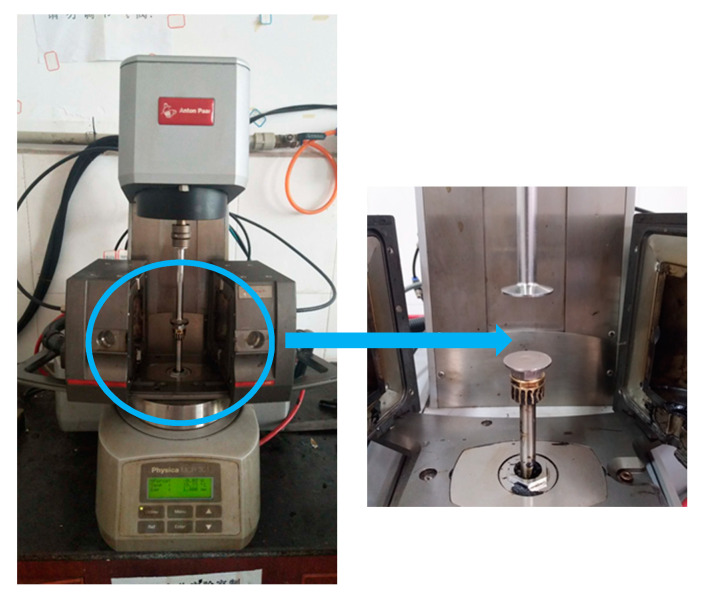
Dynamic shear rheometer (DSR) test equipment.

**Figure 3 materials-13-04437-f003:**
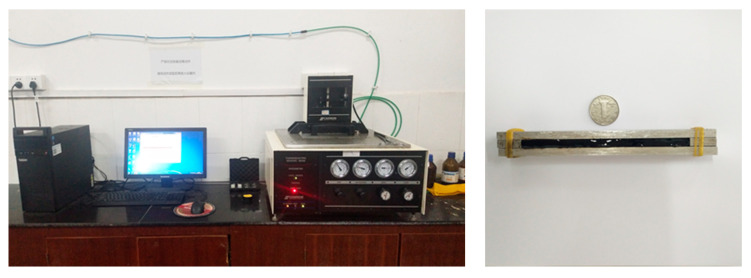
Beam bheometer (BBR) test equipment.

**Figure 4 materials-13-04437-f004:**
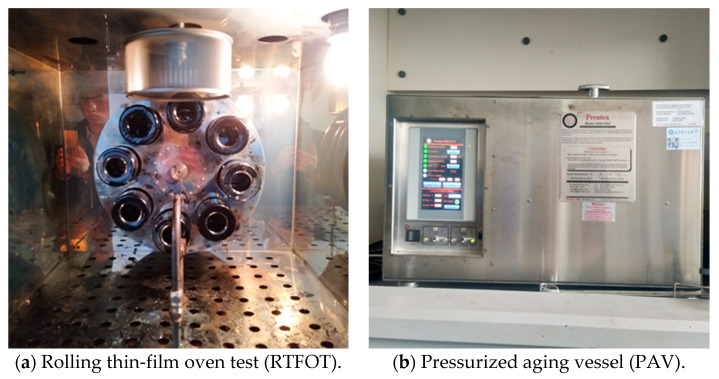
Aging test equipment.

**Figure 5 materials-13-04437-f005:**
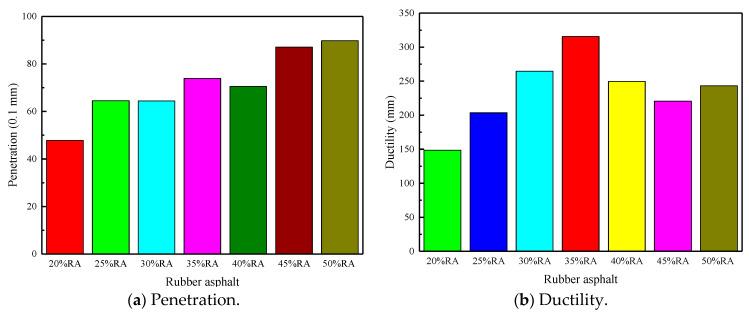
Penetration, ductility, softening point, and rotation viscosity.

**Figure 6 materials-13-04437-f006:**
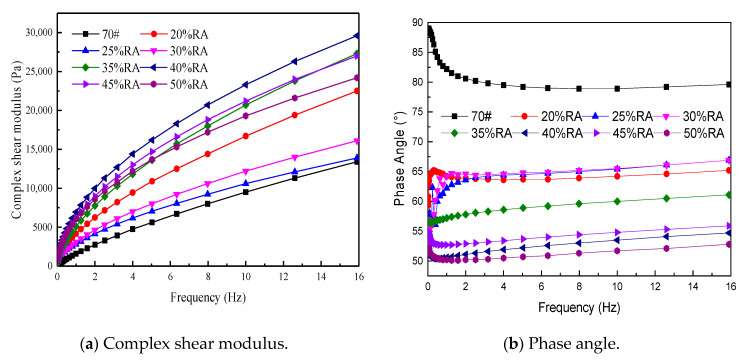
Complex shear modulus and phase angle. (**a**) Complex shear modulus (**b**) Phase angle

**Figure 7 materials-13-04437-f007:**
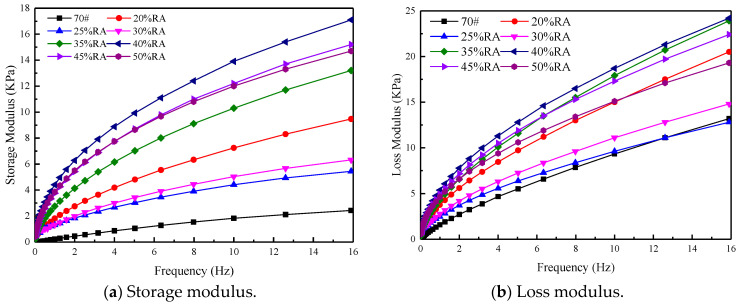
Storage modulus and loss modulus. (**a**) Storage modulus (**b**) Loss modulus.

**Figure 8 materials-13-04437-f008:**
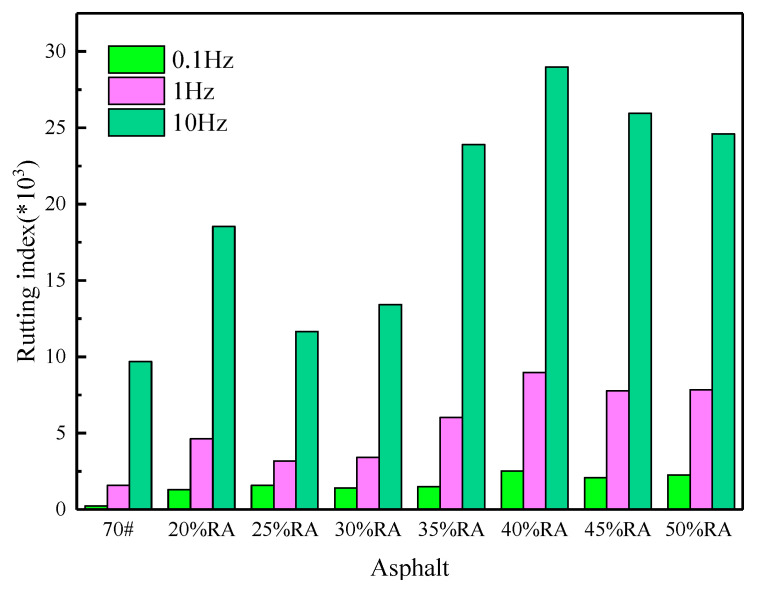
Rutting index of high-content rubberized asphalt.

**Figure 9 materials-13-04437-f009:**
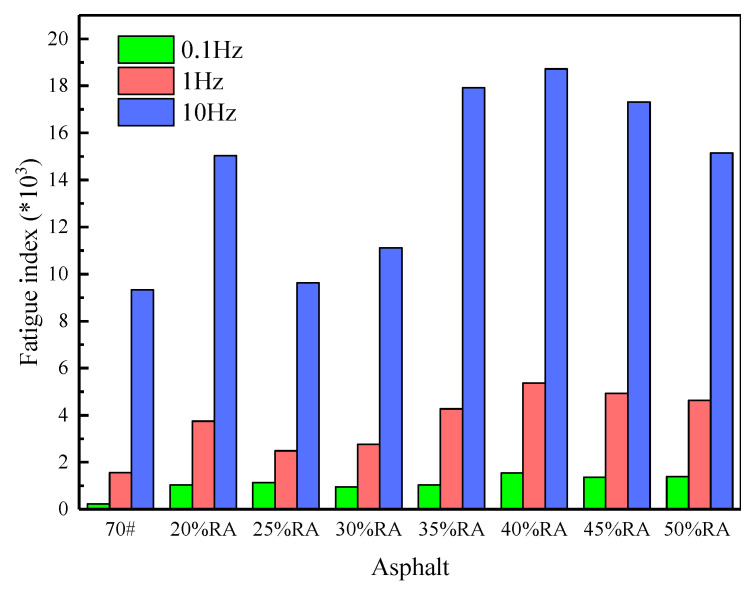
Fatigue index of high-content rubberized asphalt.

**Figure 10 materials-13-04437-f010:**
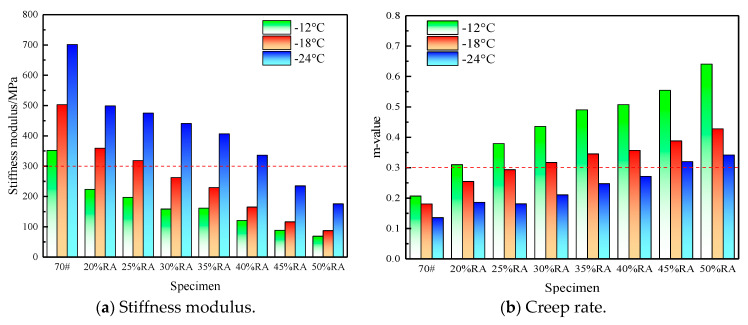
Low-temperature stiffness modulus and creep rate of high-content rubberized asphalt. (**a**) Stiffness modulus (**b**) Creep rate.

**Figure 11 materials-13-04437-f011:**
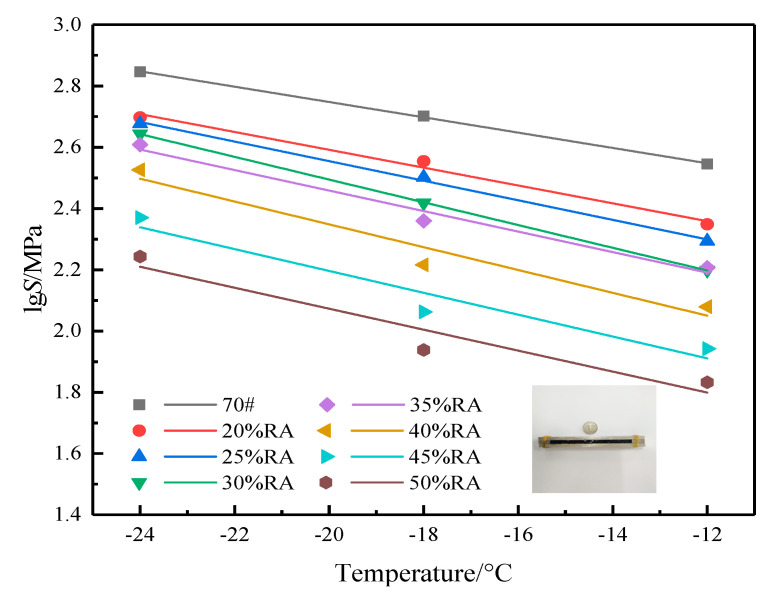
Low-temperature sensitivity of high-content rubberized asphalt.

**Figure 12 materials-13-04437-f012:**
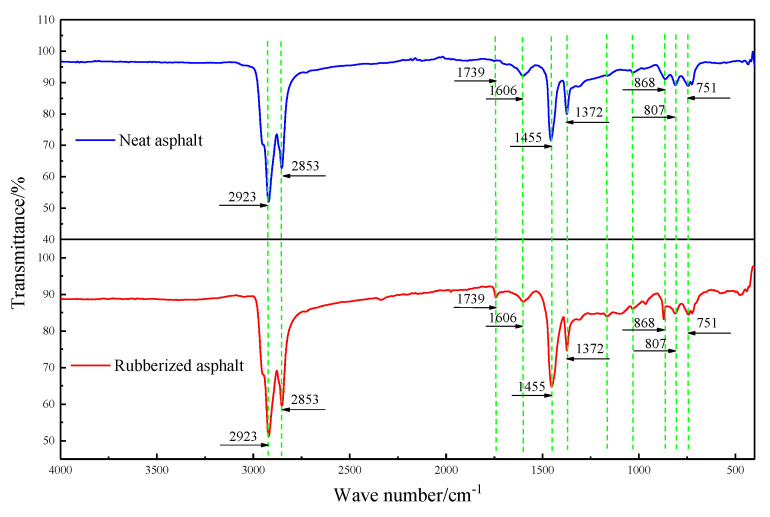
Modification mechanism of high-content rubberized asphalt.

**Figure 13 materials-13-04437-f013:**
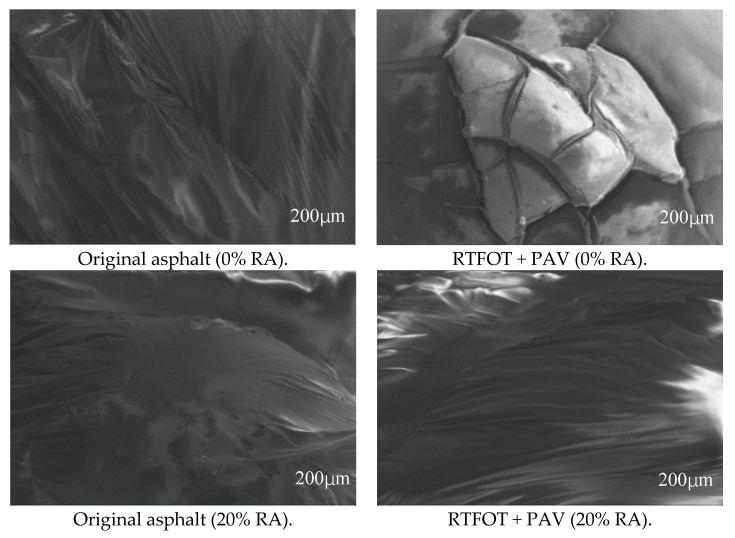
Microstructure of high-content rubberized asphalt.

**Table 1 materials-13-04437-t001:** Indices of neat asphalt.

Index	Parallel Test			Average Value	Technical Requirement
	1	2	3		
Penetration/0.1 mm	64.1	64.5	67.6	65.4	60–80
Ductility (15 °C, 5 cm/min)/cm	133.1	133.3	133.7	133.4	≥100
Softening point/°C	49.5	50.2	49.85	49.9	≥46
Dynamic viscosity (60 °C)/Pa·s	198.2	189.5	205.1	197.6	≥180
Wax content/%	2.08	1.99	1.87	1.98	≤2.2
Density (15 °C)g/cm^3^	1.021	1.025	1.02	1.022	–

**Table 2 materials-13-04437-t002:** Properties of rubber powder.

Type	Ash Content/%	Acetone Extract/%	Carbon Black Content/%	Rubber Hydrocarbon Content/%	Tensile Strength/MPa	Elongation at Break/%
Rubber powder	6	9.21	32	55	16	620

**Table 3 materials-13-04437-t003:** Fitting results of low-temperature sensitivity analysis.

Asphalt Binder	*S_TS_*	*C*	*R* ^2^
70#	−0.025	2.247	0.999
20%	−0.0291	2.009	0.978
25%	−0.0319	1.915	0.994
30%	−0.0371	1.752	0.999
35%	−0.0334	1.789	0.963
40%	−0.0372	1.603	0.904
45%	−0.0356	1.482	0.88
50%	−0.0342	1.388	0.853

**Table 4 materials-13-04437-t004:** Economic analysis of rubberized asphalt.

Rubberized Asphalt	20% RA	25% RA	30% RA	35% RA	40% RA	45% RA	50% RA
Rubber powder content (%)	20	25	30	35	40	45	50
Neat asphalt content (%)	80	75	70	65	60	55	50
Unit price of rubber powder (RMB/ton)	1500	1500	1500	1500	1500	1500	1500
Unit price of neat asphalt (RMB/ton)	2900	2900	2900	2900	2900	2900	2900
Unit price of rubber powder and neat asphalt (RMB/ton)	2620	2550	2480	2410	2340	2270	2200
Admixtures and labor costs (RMB/ton)	800	900	950	1000	1100	1150	1200
Total cost (RMB/ton)	3420	3450	3430	3410	3440	3420	3400

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
