# Peer review of "Laboratory Investigation of Rubberized Asphalt Using High-Content Rubber Powder"

_materials, 2020, doi:10.3390/ma13194437_

Round 1
Reviewer 1 Report
The article looks good, but unfortunately there are issues that need serious discussion and criticism. Pavement engineers have long sought to increase the amount of waste rubber in bitumen to take advantage of its environmental benefits, but have not succeeded or research is still ongoing. But what this research suggests could be interesting, if viscosity testing was reported in this article. Unfortunately, the big downside of this paper is the lack of a reported bitumen viscosity test.
Nonetheless, the method of preparation of modified bitumen is very vague and the various materials used to facilitate the mixing process of rubber and bitumen are still vague (i.e. viscosity reducing agent, stabilizer etc.), thus this article cannot be easily judged at all.
There are also several writing problems throughout the article, for example sentences 471 to 474.
The economics of using large amounts of rubber in bitumen have not been investigated. The question is whether doing all this mixing process and using different agent materials still justifies the economical benefits of using rubber?
Since there are fundamental questions as well as important ambiguities in this article and the writing quality is not appropriate.
Reviewer 2 Report
Dear authors
Firstly I would like to congratulate for interesting and useful studies. However I have following remarks and please consider them to the final version of your paper:
Describe in more details the process of mixing rubber powder with neat bitumen, present some photo of the pilot reactor. What are the possibilities to apply analogous process in industry? Does special installation required? Where the crumb rubber would be add, in a refinery or in an asphalt mixture plant? Please include the answers in your text.
Figure 5 a) what does green line mean? Figure 5 b) x-axle should be “Frequency”
I recommend to perform in future works MSCR test to evaluate anti-rutting performance. There are several works that prove MSCR is more appropriate than G*/sin delta.
Similarly LAS test is more appropriate to evaluate fatigue resistance of asphalt bitumen. I do not see the necessity to make an additional tests at this moment, but some comment can be add in the text.
Summarize what is the major benefit of application crumb rubber powder. In my opinion the most visible and obvious improvement is visible in low-temperature properties.
You can present how the PG grades of neat and RA modified bitumen change with the increase RA content.
Add some comment how the improvement of properties of the RA modified bitumen correspond to other modifiers, like SBS or EVA which are commonly used.
Line 473/p16, should be “the” not “thee”
Reviewer 3 Report
Thank you for submitting this paper.
This is an interesting work on a current topic. The paper shows a detailed rheological analysis on rubberized asphalt mixtures containing high amount of rubber.
The paper is well written, well organized and easy to follow. The results are in line with the conclusions presented by the authors.
I would suggest working on few minor changes:
- Line 51. According to the authors, ordinary crumb rubber is incompatible with asphalt. Despite the bibliographic reference, this statement could easily be refuted. Several remarkable studies demonstrate the opposite. I would suggest modifying the authors’ statement.
- Units of measure should be presented in brackets.
- The letters of the different figures (i.e. a, b and c) should be explained in the figures’ captions.
- Line 256. Could the authors clarify the cited ductility requirements?
- Lines 275-278. It this explanation needed?
- Figure 5. Could the authors include the Complex shear modulus and the phase angle showed in Figure 5a into Figure 5b and 5c? You did the same for Figure 6 and I think this could improve the readability of the Figure and help the reader to compare the results.
- According to the results showed in Figure 5, 6 and 7, the 20%RA does not follow a clear trend. How can the authors explain the decrease in the rutting index from 20%RA to 25 %RA? The same phenomenon is visible in Figure 5 and 6 in which 20%RA performs better if compared to 25 and 30%RA.
Reviewer 4 Report
Please address my comments:
Title: it seems to me more clear like following: “Laboratory Investigation of Rubberized Asphalt Using High Content of Rubber Powder”
Abstract: The abstract should be a total of about 200 words maximum, according to Instructions for authors. Yours has 339 words.
SI Units: (International System of Units) should be used. Imperial, US customary and other units should be converted to SI units. Example: page 1, line 16 - (…) 30 mesh waste rubber (…)
Page 1, Section 1 (Introduction): I believe that this section is too long, written with a "commercial spirit" and not pretty much informative. RA is to be used in asphalt mixtures (AM), and you didn't mention the present way in the world technology of producing and using crumb rubber AM. It should be related to these present techniques, the comparison of using a high content of rubber RA. Even if you want to stay in the Asphalt cement field and laboratory, you should state this fact and discuss your proposal having as reference the typical crumb rubber RA. So, I believe that you can be sharper and more consistent in this first section for the readers' sake.
Page 8, Line 250: Figures and Tables should make their appearance in the article after being mentioned in the main text. Was not the case for Figure 4, for example. It is a question of format but should be respected.
Page 15, Section 5 (Conclusions): The authors need to strengthen this section or introduce a section of Discussion to lead the insights about the laboratory results obtained. Several issues should be assessed as why HCRA with 40% to 50%? Why not 35% or even 30%? What should be expected in terms of the production of asphalt mixtures? How to do it? Is this a problem? Are the results in line with others, from other researchers, for 20% of CR incorporation? Why 0,6mm size for the rubber? Why not other dimensions? Are the overall application costs expected to be what? Much higher, moderate-higher, what? This section should not be a list of results of even worst a guess from not completely clear results, and application in asphalt mixtures should be briefly discussed to graduate the type of aimed enhancement with HCRA.
Round 2
Reviewer 1 Report
I find the changes and responses satisfactory.
Reviewer 4 Report
In my view, the authors fairly addressed my comments. Despite some issues remained not completely understood by the authors, I believe that the article is now in fair condition to be published. A final remark to state that the authors miss the particle "of" in the title:
Laboratory Investigation of Rubberized Asphalt Using High Content of Rubber Powder